# BET Protein-Mediated Transcriptional Regulation in Heart Failure

**DOI:** 10.3390/ijms22116059

**Published:** 2021-06-04

**Authors:** Talha Ijaz, Michael A. Burke

**Affiliations:** Division of Cardiology, Department of Internal Medicine, Emory University School of Medicine, Atlanta, GA 30322, USA; Talha.Ijaz@emory.edu

**Keywords:** BET, BRD4, heart failure, dilated cardiomyopathy, phospholamban, chromatin remodeling, superenhancer, transcription regulation

## Abstract

Heart failure is a complex disease process with underlying aberrations in neurohormonal systems that promote dysregulated cellular signaling and gene transcription. Over the past 10 years, the advent of small-molecule inhibitors that target transcriptional machinery has demonstrated the importance of the bromodomain and extraterminal (BET) family of epigenetic reader proteins in regulating gene transcription in multiple mouse models of cardiomyopathy. BETs bind to acetylated histone tails and transcription factors to integrate disparate stress signaling networks into a defined gene expression program. Under myocardial stress, BRD4, a BET family member, is recruited to superenhancers and promoter regions of inflammatory and profibrotic genes to promote transcription elongation. Whole-transcriptome analysis of BET-dependent gene networks suggests a major role of nuclear-factor kappa b and transforming growth factor-beta in the development of cardiac fibrosis and systolic dysfunction. Recent investigations also suggest a prominent role of BRD4 in maintaining cardiomyocyte mitochondrial respiration under basal conditions. In this review, we summarize the data from preclinical heart failure studies that explore the role of BET-regulated transcriptional mechanisms and delve into landmark studies that define BET bromodomain-independent processes involved in cardiac homeostasis.

## 1. Introduction

Heart failure (HF) is the end result of the pathological remodeling of the heart, which results in a supply/demand mismatch and consequent decrease in perfusion to vital organs. A multitude of factors can promote pathological remodeling, including chronic hypertension, myocardial infarction (MI), viral infections, and genetic mutations, that affect the contractile function of cardiomyocytes [1,2,3]. With unabated stress, the myocardium undergoes pathological remodeling that typically includes myocyte hypertrophy, increased cardiac fibrosis, maladaptive Ca^2+^-cycling, and, eventually, systolic dysfunction [3]. This negative remodeling of the heart predisposes individuals to lethal arrhythmias, sudden death, and the development of overt HF.

Animal and clinical studies have verified that overactivation of the renin–angiotensin system (RAAS) and β-adrenergic receptor signaling underlie the development of hypertrophy and negative cardiac remodeling [4]. Treatment with inhibitors of the RAAS and β-blockers can halt negative remodeling and, in some cases, reverse cardiac hypertrophy and ventricular chamber dilation, conferring significant mortality benefit. Mechanistically, these inhibitors limit neuroendocrine signaling cascades that promote negative remodeling. This detrimental neuroendocrine signaling activates multiple intracellular signaling pathways, including cAMP/PKA/PLN and PLC/Ca^2+^-calmodulin, that result in the binding of transcription factors (TFs) such as NFAT, MEF2, NF-κB, and GATA4 to DNA, thus promoting the transcription of genes necessary for pathological remodeling (Figure 1) [1]. Although multiple signaling cascades are activated under cardiac stress, evidence gathered over the past few years suggests that certain common epigenetic systems regulate myocardial gene transcription across the entire HF spectrum. Thus, targeting such mechanisms for therapeutic benefit holds great promise for the treatment of HF.

Gene expression is controlled by DNA binding TFs and chromatin changes that allow the transcriptional machinery to be recruited to target genes. The binding of TFs to DNA elements known as enhancers stimulates the recruitment of other cofactors and RNA polymerase II (pol II) to the transcription start site [5]. Usually, multiple enhancers span across thousands of base pairs necessitating the looping of DNA to promote cooperative interplay between DNA-binding proteins and pol II. Essential cofactors include enzymes that modify histones by post-translational modifications, including acetylation, methylation, phosphorylation, and ubiquitination. Histone modifications can be recognized by epigenetic “reader” proteins that bind to acetylated histone residues to promote protein complex formation on the gene promoter and are a crucial part of the transcription machinery. Specific histone modifications correlate with transcriptional activity; for instance, acetylation of lysine residue 27 on histone H3 (H3K27ac) and trimethylation at histone H3 lysine residue 36 (H3K36me3) are found on promoters and enhancers of active genes [6,7]. By contrast, trimethylation of histone H3 lysine 27 (H3K27me3) correlates with transcriptional repression [8]. While enhancers are specific regulatory regions composed of clusters of TF-binding motifs, so-called super-enhancers (SEs) are clusters of enhancers with very high levels of H3K27ac marks and master TF occupancy [9]. Recruitment of the transcriptional machinery to these regulatory regions is essential for robust, dynamic changes in gene expression. Once pol II initiates transcription, it pauses after 20–50 nucleotides generating a nascent mRNA. Release of the paused state requires recruitment of the positive transcription elongation factor b (P-TEFb) complex, which phosphorylates pol II at serine-2 to promote transcription elongation of mature mRNA transcripts [10,11,12].

Histone acetylation, in particular, is an important component of cardiac remodeling in the progression to HF. ChIP-Seq analysis of cardiomyocytes from hypertrophied hearts has indicated that nearly 10% of the genome undergoes changes in the distribution of histone marks during stress, with all histone marks being redistributed to transcription start site [13]. The differential distribution of H3K9ac and H3K27ac marks were mapped to the regulatory domains of genes involved in maintaining cardiac function or in the regulation of gene expression. When correlated with RNA-seq data, the decrease in H3K9ac or H3K27ac was associated with transcriptional repression whereas increase in these marks was associated with increase gene expression, highlighting their role as signals for the recruitment of transcriptional machinery [13,14]. H3K9ac marks also directly correlates with pol II activity, with an increase in H3K9ac marks localizing to regions of increased pol II occupancy during pathological hypertrophy [15]. These studies highlight the key role of histone acetylation in mediating gene transcription during the development of HF.

Further evidence of the involvement of histone acetylation in HF comes from studies investigating histone acetyl transferases (HATs) and histone deacetylases (HDACs), which add or remove acetyl-lysine groups from histone side chains. The HAT p300 is a transcriptional coactivator that increases during cardiac hypertrophy [16]. Haploinsufficiency of *Ep300* prevented mice from developing hypertrophy in response to pressure-overload, whereas overexpression of p300 led to pathological remodeling associated with hyperacetylation of chromatin and pro-hypertrophic cardiac transcription factors [16]. Similarly, inhibition of class I and II HDACs prevents pathological hypertrophy in response to multiple stress stimuli [17,18,19]. Discussion of HDACs in HF have been reviewed in-depth elsewhere [20]. Collectively, these studies demonstrate that histone acetylation, particularly of lysine residues, is a dynamic process that has a profound impact on cardiac remodeling in response to stress. 

Acetylated lysine residues are specifically recognized by bromodomains (BDs), which are highly conserved protein motifs found in many chromatin-associated proteins [21]. BDs are found in nearly 50 different proteins that can be grouped into 8 families [22]. The BD and extraterminal (ET) domain containing (BET) family consists of BRD2, BRD3, BRD4 and testis-specific BRDt. BET proteins recognize acetyl-lysine residues on chromatin tails and other proteins including acetylated-TFs via their BD domains [23,24]. All four family members contain two N-terminal BD domains and a C-terminal ET domain (Figure 2). BRD4 is the most well studied of the family; it contains a C-terminal region that interacts with P-TEFb and promotes transcription elongation [14]. Protein levels of BRD2, BRD3 and BRD4 are similar in the LV myocardium with no significant difference in the gene expression of these three family members between males and females [25]. BRDt is only expressed in the testes. In the heart and other tissues, BRD4 has a well-established role in transcription elongation, while little is known about BRD2 or BRD3 in the myocardium [14]. In Th17 immune cells, BRD2 interacts with the chromatin insulator CTCF to facilitate SE formation, BRD4 recruitment and gene expression changes via direct BRD2-BRD4 interaction [26]. BRD3 appears to regulate rRNA and ribosomal biogenesis in human sarcoma cells, thereby exerting effect on cellular proliferation [27]. In this review, we aim to describe the role of BET proteins, specifically BRD4, in transcriptional regulation during the development of HF.

## 2. BET Proteins in Transcriptional Regulation

BET proteins function as epigenetic readers of acetylated histone marks and a scaffold to localize the transcriptional machinery at the transcription start site (Figure 2). The interaction between BRD4 and P-TEFb is necessary for the phosphorylation of serine-2 on pol II to release it from its paused state and thereby permit full transcription of the gene [10]. This critical interaction can be disrupted by treatment with BET inhibitors, thereby preventing dynamic changes in gene expression. Interestingly, selective degradation of BET proteins does not alter the binding of CKD9 to DNA, but still completely blocks pol II phosphorylation and transcriptional elongation, highlighting the central importance of BRD4 to this process [28]. Additionally, BRD4 can function as an atypical kinase that has the ability to phosphorylate pol II serine-2 to induce transcriptional elongation [29]. Therefore, BRD4 is a critical nodal protein for stimulus-coupled P-TEF-b activation and the pause-release of pol II, which drives dynamic changes in transcription. 

BETs also regulate gene expression by associating with the Mediator complex, a multimeric protein complex that links TFs and pol II [30]. It remains unclear if BRD4 directly binds to the Mediator complex, but BRD4 localizes with the Mediator complex at specific genomic locations, including SE, suggesting that it stabilizes the transcriptional apparatus. In addition, histones H3 and H4 can be directly acetylated by BRD4, thus demonstrating intrinsic histone acetyl transferase (HAT) activity and signifying another mechanism for regulating transcription [31]. H3K122 acetylation by BRD4 leads to chromatin decompaction, allowing for an increase in target gene transcription [31]. Further evidence suggests that BRD4 HAT activity is needed for the induction of transcription, as blockade of BRD4 with a small molecule can decrease H3K122 acetylation and BRD4-dependent inflammatory gene expression [32]. More recent evidence suggests that BRD4 is involved in regulating RNA splicing via its HAT activity and its ET domain. BRD4 interacts with the RNA splicing regulator HnRNPM, and with alternative exons to promote transcription of splice variants [33].

The presence of BRD4 at SEs, in combination with the Mediator complex, helps to compartmentalize the transcription machinery. BRD4 and MED1 (a component of the Mediator complex) localize in nuclear bodies and occupy SEs in embryonic stem cells [34]. These nuclear bodies behave like phase-separated condensates to form transcriptional compartments and aid in the robust transcription of genes involved in maintaining cell identity. BRD4 has also been identified as a key transcriptional co-activator in cellular differentiation during adipogenesis and myogenesis. Lineage-determining TFs on active enhancers recruit BRD4 in coordination with the H3K4 methlytransferases MLL3 and MLL4, and the H3K27 acetyltransferases CBP/p300, to enhancers [35]. Deletion of BRD4 prevented adipocyte differentiation by inhibiting enrichment of Mediator and pol II machinery but not lineage-determining TFs, H3K4me1 marks or H3K27ac marks on enhancers. In contrast to its critical role in maintaining stem cell identity, BRD4 was not necessary to maintain adipocyte cell identity after differentiation.

Gene transcription is also facilitated by the switching/sucrose non-fermentable (SWI/SNF) complex that rearranges nucleosome positions thereby permitting the access of transcription machinery to target genes. BRD4 pairs with BRG1, a component of SWI/SNF, to increase transcription at pluripotency genes in embryonic stem cells [36]. Nucleosome remodeling mediated by BRD4-BRG1 maintains cellular phenotype and can aid in differentiation. Within the myocardium, BRG1 promotes embryonic myocyte proliferation and preserves fetal cardiac differentiation [37]. BRG1 is turned off in adult cardiomyocytes but can be reactivated under cardiac stress to promote an embryonic fetal program. 

Understanding the role of BET proteins in the regulation of gene transcription has been aided by the development of small molecule-inhibitor that actively compete with acetyl-lysine for BD binding. The first-in-class BD inhibitor, JQ1, was found to disrupt assembled transcriptional machinery and block gene expression [22]. The advent of JQ1 and other BD domain inhibitors, including I-BET151 and RVX-208, has allowed for the exploration of the role of BETs in the development of cardiovascular disease and HF [23]. 

## 3. BET Protein Function in Cardiomyocytes In Vitro

In neonatal rat ventricular myocytes (NRVMs), BET inhibition with JQ1 prevented cardiomyocyte hypertrophy, and blunted the differential expression of hypertrophy-associated genes such as *Anf*, *Bnp*, and *Serca2a* [14,38]. Chromatin immunoprecipitation (ChIP) analysis of the transcriptional start site of *Anf* further clarified that JQ1 is effective at disrupting the transcriptional machinery at specific loci in cardiomyocytes. Further, BRD4 is upregulated in NRVM without changes in *Brd4* mRNA levels under hypertrophic stress [39]. Evaluation of the 3′-untranslated region of *Brd4* identified multiple microRNA (miR) binding sites. With phenylephrine treatment (a pro-hypertrophic stimulus), miR-9 expression levels decreased suggesting that this particular miR is involved in regulating *Brd4* at the post-transcriptional level. Indeed, a miR-9 mimic suppressed phenylephrine-induced hypertrophy and BRD4 expression with transcriptomic changes that were largely similar to those observed with JQ1. Genome-wide binding of BRD4 identified 459 SEs, with increased BRD4 binding to 65 SEs and a concomitant decrease observed at 19 SEs under hypertrophic stimuli. ChIP-seq targeting RNA pol II also elucidated three subsets of promoter regions that had either increased, decreased, or unchanged the binding of BRD4. The miR-9 mimic strongly suppressed BRD4 accumulation on SEs and promoters confirming its key role in BRD4 expression and BRD4 dependent transcriptional regulation. Finally, the AP-1 transcription factor complex was identified as a key TF that binds to BRD4-enriched SEs to promote gene transcription. Collectively, these landmark studies in cardiomyocytes demonstrated that (1) BETs are involved in cardiomyocyte hypertrophic gene expression; (2) hypertrophic signaling promotes a shift in BRD4 loading on SEs, which facilitates gene transcription; and (3) miR-9 is an important regulator of BRD4 expression.

In addition to JQ1, other BET inhibitors including I-BET, I-BET-151, RVX-208, and PFI-1 were successful at inhibiting cardiomyocyte hypertrophy, suggesting that this is a class effect of BET inhibitors [14]. RNA-seq assessment in cardiomyocytes after phenylephrine stimulation identified the induction of several hundred genes, many of which were suppressed by JQ1. Functional pathway analysis suggested that BETs were involved in cytoskeletal reorganization, ECM production and proinflammatory signaling. These studies provided a strong basis for examining the role of BET inhibition in vivo.

## 4. BET Involvement in Pressure-Overload Cardiomyopathy

Transverse aortic constriction (TAC) leads to LV pressure overload and serves as a model of acute HF. Daily JQ1 administration immediately following TAC surgery successfully inhibited cardiac hypertrophy and LV systolic dysfunction for up to 4 weeks [14,38]. Likewise, JQ1 successfully blocked phenylephrine-induced cardiac remodeling over a similar period of time [14]. BET inhibition was associated with decreased cardiac fibrosis and apoptotic cell death. RNA-seq on whole LV tissue samples revealed that JQ1-mediated changes could be observed as early as 3 days and were separated into three different clusters: TAC-induced and JQ1-suppressed, TAC-induced and unaffected by JQ1, and TAC-suppressed. This suggests that BET inhibition only blocks a subset of stress-induced genes, yet this is still effective at limiting pathologic cardiac remodeling. Functional pathway analysis grouped most of the JQ1-suppressed genes into those involved in ECM remodeling and inflammation.

Gene set enrichment analysis (GSEA) was performed to compare JQ1-induced gene changes with previously established clusters of genes known to be regulated by specific TFs including calcineurin-A, NF-κB, and GATA4 [14]. This revealed that TAC-induced transcriptional profiles were enriched for genes that are dependent on calcineurin-A, NF-κB and GATA4, and JQ1 promoted negative enrichment of these TF signatures. ChIP-seq performed on whole LV tissue demonstrated that BRD4 was enriched at enhancer elements identified by H3K27ac marks and promoters of actively transcribed genes identified by H3K4me3 marks. BET inhibition attenuated pol II occupancy in regions of transcriptional elongation and enhanced its occupation at promoters, thereby indicating that BET proteins activate gene transcription by controlling the pol II pause-release mechanism.

In a follow-up study to examine whether BET inhibition could effectively treat established cardiomyopathy, a cohort of mice that underwent TAC were treated with JQ1 2.5-weeks after surgery [40]. JQ1 successfully attenuated systolic dysfunction and limited cardiac hypertrophy verifying that BET inhibition is effective at limiting myocardial damage after cardiomyopathy has been established, an important clinical correlate for HF, which is usually well-established before being detected in patients. 

## 5. Role of BETs in Ischemic Cardiomyopathy

In a model of MI caused by proximal LAD occlusion, BET inhibition with JQ1 suppressed the development of cardiomyocyte hypertrophy, LV systolic dysfunction and cavity dilation [40]. Further, JQ1 treatment limited fibrosis in regions of the LV remote from the site of infarction. Transcriptomic profiles showed significant overlap between stress inducible genes that were suppressed by JQ1 in both MI and TAC models and could be grouped into processes for the regulation of ECM deposition, inflammatory response and cellular growth. Pathway analysis revealed enrichment for transforming growth factor β (TGFβ)-signaling and immune response pathways that converge on NF-κB, AP-1, and STAT1 between JQ1- and vehicle-treated mice. To further understand the cell-specific effects, RNA-seq profiles from whole LV were compared to previously curated data from mouse cardiomyocytes, mouse cardiac fibroblast and myeloid cells. Bioinformatics analysis indicated the activation of fibroblast genes in a TAC/MI model which underlies the development of HF. 

In contrast to the importance of BETs in pathological stress-induced cardiomyopathy, BET inhibition by JQ1 had no effect on physiological hypertrophy. Adult mice subjected to a high-intensity endurance swimming protocol for 3 h/day for 4 weeks displayed mild cardiomyocyte hypertrophy with a concomitant increase in LV systolic function irrespective of treatment with JQ1 [40]. A key difference between physiologic and pathologic hypertrophy is the activation of inflammatory gene networks, which are a hallmark of the latter. Therefore, this critical result demonstrates that BETs are not involved in regulating exercise induced cardiac hypertrophy but function specifically in regulating pathological stress-induced gene transcription.

## 6. BETs in Genetic Dilated Cardiomyopathy

Genetic mutations in the Ca^2+^-regulatory gene *phspholamban* (*PLN*) lead to the development of dilated cardiomyopathy (DCM) characterized by LV chamber dilation and HF. Our lab previously characterized transgenic mice overexpressing a specific human *PLN* mutation (Arg9Cys; PLN^R9C^), that causes HF and early death within 5–6 months [41]. PLN^R9C^ mice with DCM exhibit significant myocardial fibrosis associated with increased fibroblast proliferation. To discern cell-type specific changes, RNA-seq was performed after the separation of cardiomyocytes from non-myocytes (i.e., fibroblasts, endothelial cells, leukocytes, etc.); this identified the upregulation of profibrotic cytokines including TGFβ2, TGFβ3, and CTGF in cardiomyocytes, whereas periostin and components of NF-κB signaling were upregulated in the non-myocyte population [41]. GSEA of enriched genes in PLN hearts highly correlated with genes that are suppressed by JQ1 in TAC and MI models, suggesting a role of BET proteins in mediating PLN-DCM. 

Treatment of PLN^R9C^ mice with JQ1 inhibited systolic HF, cardiac fibrosis, and fibroblast proliferation and increased the lifespan by 10%, thereby verifying the role of BETs in genetic DCM [25]. RNA-seq assessment of cellular compartments indicated that JQ1 treatment induced profound changes in cardiac nonmyocytes. Specifically, BET inhibition uniformly suppressed inflammatory gene networks that were activated in PLN^R9C^ nonmyocytes with DCM. In contrast, the cellular metabolic changes observed in cardiomyocytes remained unperturbed by JQ1 treatment. Mechanistically, acetylated RelA (aK310 –RelA) was markedly increased in PLN hearts and was identified to be bound to BRD4. JQ1 blocked this RelA-BRD4 interaction while having no effect on aK310-RelA levels, thus suggesting that NF-κB is still activated, but BRD4 is necessary for the expression of at least a subset of NF-κB-activated genes in PLN^R9C^ hearts. 

Lamin A/C (LMNA) is a component of the nuclear envelope and genetic mutations in *LMNA* account for approximately 6% of familial DCM cases [2]. In an animal model of LMNA cardiomyopathy, mice developed DCM, cardiac arrhythmias and myocardial fibrosis [42]. Similar to the transcriptomic changes observed in the heart with TAC, MI and PLN^R9C^, RNA-seq analysis of LMNA-DCM hearts identified the upregulation of genes involved in inflammation, epithelial-to-mesenchymal transition (EMT), and apoptosis, with TGFβ1, RelA/NF-κB, and STAT1 pathways being highly activated. ChIP-seq analysis of BRD4 revealed that BRD4 peaks were redistributed in cardiomyocytes from LMNA-DCM mice and localized with genes that have increased transcript levels compared to WT cardiomyocytes. BET inhibition with JQ1 increased median survival from 23 to 32 days, limited negative LV remodeling, decreased cardiac arrhythmias, and markedly inhibited fibrosis and cell death. BET inhibition was associated with suppression of the innate immune response, EMT, and fibrosis-associated gene expression.

## 7. BRD4 Mediated Regulation of Cardiac Homeostasis

Although numerous studies utilized JQ1 to demonstrate the vital role of BETs in inflammatory and fibrotic gene expression during HF development, it remains to be defined whether BRD4 was the dominant BET protein mediating transcriptional changes. To assess this, two separate groups created cardiomyocyte-specific BRD4-null mice and made similar observations. Cardiomyocyte-specific deletion of BRD4 in early gestation was embryonically lethal by postnatal Day 5 [43]. In contrast, later deletion of BRD4 (~e12), did not lead to cardiac defects at birth but led to progressive contractile dysfunction, pulmonary edema (a sign of HF), and nearly 100% mortality within 10 weeks [44]. BRD4 deletion was associated with decreased LV mass and eccentric remodeling. Similarly, cardiomyocyte-BRD4 deletion in adult mice also led to progressive systolic dysfunction, LV dilation, and increased mortality [43,44].

RNA-seq analysis of whole LV between JQ1-treated and BRD4-null hearts identified enrichment in genes associated with mitochondrial metabolism and the electron transport chain (ETC) pathway [44]. Mitochondria isolated from BRD4-null hearts demonstrated marked deterioration of ETC activity involving complex I, III, and IV and a decrease in TCA cycle enzymes. Further, mitochondria from BRD4-null hearts reduced oxygen consumption in the presence of fatty acid and pyruvate substrates. Changes in cardiomyocyte mitochondrial metabolism were linked to the downregulation of PGC1α and PGC1β, known master regulators of mitochondrial metabolism [43]. Furthermore, ChIP-seq analysis indicated co-occupying signals from H3K27ac/BRD4/GATA4 at 28% of dysregulated genes in BRD4-null hearts, including on enhancer elements at the PGC1α locus. Biochemical analysis verified that BRD4 interacts with GATA4, but surprisingly, in a BD-independent manner [43]. Collectively, these studies identified an important role of BRD4 in maintaining cardiomyocyte mitochondrial homeostasis and energy production that occurs in a BD-independent manner.

RNA-seq analysis of whole LV between JQ1-treated and BRD4-null hearts indicated that there was only a 25% overlap of differentially expressed genes [44]. There are multiple possible reasons for these seemingly disparate results. First, JQ1 is a reversible inhibitor of BD domains and does not affect BET expression. Second, JQ1 has a short half-life of ~1hr after IV injection [22]. Therefore, in JQ1 treated animals, BETs continue to function at some level in absence of complete inhibition. Third, as a pan-BET inhibitor, JQ1 also inhibits BRD2 and BRD3, which may influence cardiac remodeling and HF. Currently, the role of these other BET family members in cardiac pathology remains undefined and is an important area of future investigation in the field. 

Another important consideration is that the non-BD functions of BETs have a wide range of effects, some of which are vital to maintaining normal cell functions [45]. Hence, complete ablation as in genetic knockout or protein degradation studies will have different effects than BD-specific inhibition. The studies detailed above also suggest a BD-independent role of BRD4 in maintaining cardiomyocyte homeostasis. Moreover, the administration of JQ1 affects all myocardial cell types including fibroblasts, endothelial cells and resident macrophages. Thus, stress-induced regulation of gene expression by BRD4 may occur via BD-dependent and BD-independent mechanisms, with distinct responses in different cardiac cell types. Collectively, these data demonstrate the important biological differences between studies of pharmacological inhibition and genetic ablation, a critical aspect to consider when evaluating any potential therapeutic strategy.

## 8. Summary and Conclusions

BET proteins are epigenetic coactivators that are recruited during cellular stress to mediate dynamic changes in gene transcription. They integrate the vast array of pathologic signaling cascades activated in HF of varying etiologies to produce a coordinated set of gene networks that result in negative cardiac remodeling. JQ1 is a powerful inhibitor of inflammatory and fibrotic gene transcription, which helps preserve cardiac function. While JQ1 has shown therapeutic efficacy in limiting HF in preclinical models, deletion of BRD4 from cardiomyocytes is detrimental, leading to spontaneous DCM (Figure 3). Further, the function of BRD2 and BRD3 and the cell type-specific roles of different BET isoforms in the heart remains to be elucidated. 

There is significant ongoing effort to translate our current knowledge of the molecular function of BETs into clinical application. Numerous clinical trials are studying BETs as primary and adjunctive chemotherapeutics for a variety of cancers [46]. In cardiovascular diseases, phase 2 clinical trials suggested a benefit of the BET inhibitor RVX-208 (apabetalone) on plasma lipoproteins, vascular inflammation and atherosclerotic plaque formation, the major cause of ischemic cardiomyopathy [47,48]. The recently published phase 3 BETonMACE trial, which enrolled 2425 patients with type 2 diabetes and acute coronary syndrome failed to show a significant reduction in death, recurrent myocardial infarction or stroke [49]. However, a secondary analysis of this trial revealed a substantial reduction in both first hospitalization for HF (hazard ratio 0.59; 95% CI 0.38–0.94) and total hospitalizations for HF (hazard ratio 0.47; 95% CI 0.27–0.83) with apabetalone. While these findings are hypothesis generating and thus require confirmation in additional clinical studies, these data are highly exciting because they are the first data to suggest benefit with BET inhibition in HF [50]. In conclusion, while much remains to be discovered regarding the multifaceted role of BETs in cardiovascular biology and disease, the future is bright further development of BET inhibition as a successful therapeutic strategy for the treatment of HF.

## Figures and Tables

**Figure 1 ijms-22-06059-f001:**
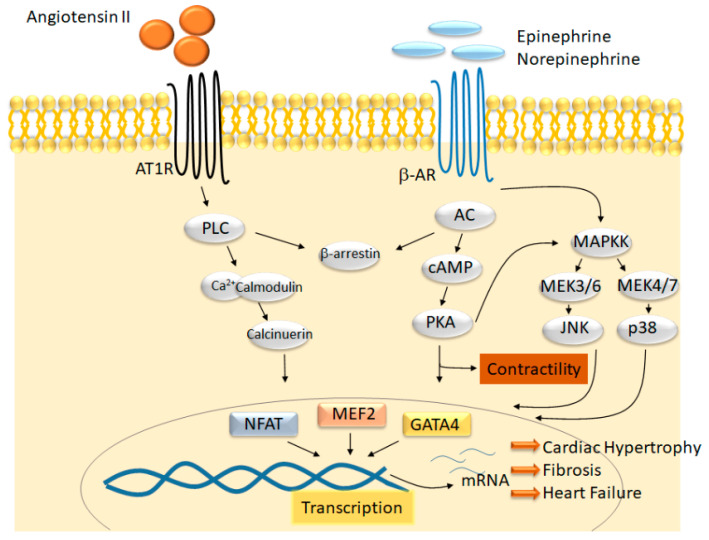
Neurohormonal signaling cascades converge on gene transcription to promote HF. Diagram of cardiac stress signaling pathways and their downstream effectors that converge on the transcriptional machinery to promote the development of cardiac hypertrophy and fibrosis, leading to HF. Signaling cascades activate well-known transcription factors, including the nuclear factor of activated T-cells (NFAT), myocyte enhancer factor 2 (MEF2), and GATA-binding protein 4 (GATA4), which are involved in pathological gene expression changes. AC, adenylyl cyclase; AT1R, angiotensin II receptor type 1; β-AR, beta-adrenergic receptor; phospholipase C; PKA, protein kinase A.

**Figure 2 ijms-22-06059-f002:**
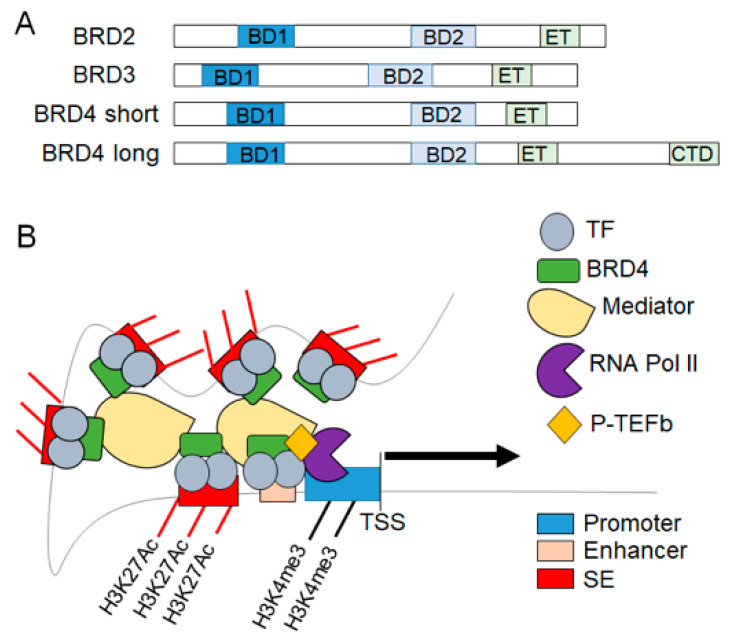
BET protein mediated transcription elongation. (**A**) The BET family of proteins contain two bromodomains (BD1, BD2). BRD4 is expressed as both a long and short isoform. BRD4 long contains a unique c-terminal domain (CTD) that interacts with P-TEFb, whereas BRD4 short lacks the CTD. (**B**) Schematic representation of the role of BRD4 in assembly of the transcriptional machinery involved in stress-induced transcription elongation of target genes in the heart. TF, transcription factor; P-TEFb, positive transcription elongation factor b; SE, superenhancer; TSS, transcription start site; red-line, H3K27Ac marks on the SE.

**Figure 3 ijms-22-06059-f003:**
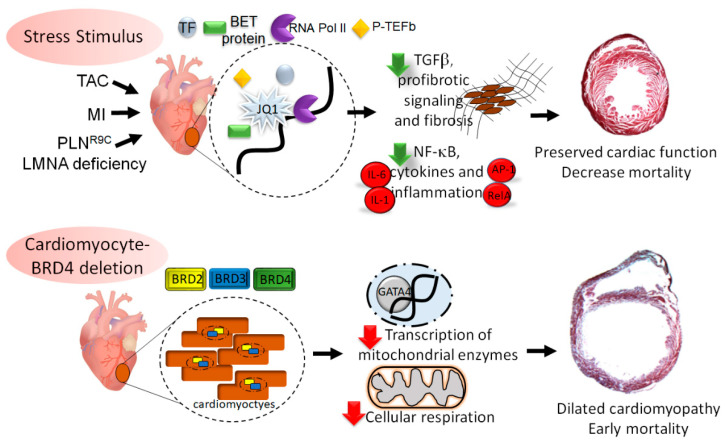
BRD4 is essential for cardiomyocyte homeostasis. Multiple forms of cardiac stress, including transverse aortic constriction (TAC), myocardial infarction (MI), and genetic defects PLN^R9C^ or LMNA deficiency could be blunted by BD inhibitor JQ1. Recent evidence also suggests that BET protein BRD4 is involved in maintaining mitochondrial homeostasis in cardiomyocyte via a BD-independent mechanism and complete deletion leads to DCM and HF.

## Data Availability

Not applicable.

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
