# Peer review of "BET Protein-Mediated Transcriptional Regulation in Heart Failure"

_ijms, 2021, doi:10.3390/ijms22116059_

Round 1

Reviewer 1 Report

The presented manuscript is a well-structured, up-to-date, and informative review about cardiac functions of BET domains. Below are few points for improvement. 

  1. Throughout the manuscript, authors should reference primary sources and not secondary reviews, for example, line 79.
  2. ET domain borders should be depicted in Figure 1.
  3. The authors could discuss how pharmacological BET inhibition targets the function of BET domains, whereas functions/interactions mediated by other domains remain largely unaffected. In contrast, genetic BRD4 ablation abolishes all possible BRD4 mediated interactions.

Author Response

We thank the reviewers for their thoughtful insights. Below is a point-by-point response to the comments provided. We are happy to address any further questions or concerns.

The presented manuscript is a well-structured, up-to-date, and informative review about cardiac functions of BET domains. Below are few points for improvement. 

  1. Throughout the manuscript, authors should reference primary sources and not secondary reviews, for example, line 79.
  2. ET domain borders should be depicted in Figure 1.
  3. The authors could discuss how pharmacological BET inhibition targets the function of BET domains, whereas functions/interactions mediated by other domains remain largely unaffected. In contrast, genetic BRD4 ablation abolishes all possible BRD4 mediated interactions.
  1. We have adjusted our references away from reviews, instead citing the primary data papers. Please note the new literature that has been cited (ref 6, 7, 11, 12, 13, 15,16,17, 18, 19, 21).
  2. We believe the reviewer is referring to figure 2, which shows the schematic layout of each BET isoform. Figure 2 has been updated as suggested.
  3. An excellent point; we have strengthened our prior discussion of this critical distinction in section 7, last paragraph (lines 340-344), where the more recent BRD4 KO studies are discussed.

Reviewer 2 Report

This review focus on the roles of BET proteins in heart failure. It summarizes the current knowledge of BET proteins, particularly BRD4, in epigenetic regulation of gene transcription in multiple mouse models of cardiomyopathy including ischemic, pressure-overload, and genetic dilated cardiomyopathy. The review also provides insight into BRD4-mediated regulation of cardiac homeostasis. The therapeutic effect of BET inhibitors and the detrimental effect of cardiomyocyte specific deletion of BRD4 are also discussed. In general, the review is well written and provides a nice overview of BET-regulated gene transcription in heart failure pathophysiology.

Minor comments:

  1. A brief overview of the expression patterns and functional characteristics of the four BRD isoforms in heart versus other tissue will be helpful.
  2. There is a typo of “to to” in line 310 of page 8.

Author Response

Reviewer 2

This review focus on the roles of BET proteins in heart failure. It summarizes the current knowledge of BET proteins, particularly BRD4, in epigenetic regulation of gene transcription in multiple mouse models of cardiomyopathy including ischemic, pressure-overload, and genetic dilated cardiomyopathy. The review also provides insight into BRD4-mediated regulation of cardiac homeostasis. The therapeutic effect of BET inhibitors and the detrimental effect of cardiomyocyte specific deletion of BRD4 are also discussed. In general, the review is well written and provides a nice overview of BET-regulated gene transcription in heart failure pathophysiology.

Minor comments:

  1. A brief overview of the expression patterns and functional characteristics of the four BRD isoforms in heart versus other tissue will be helpful.
  2. There is a typo of “to to” in line 310 of page 8.
  1. We have added a brief section discussing the known roles of the different BET isoforms in the heart and other tissues (where known; see section 1, last paragraph, lines 111-119). For clarity, we also added a preamble to that section that tells the unfamiliar reader what a bromodomain is and why it is important (lines 103-105).
  2. Noted and corrected.

Reviewer 3 Report

In this review the authors summarize the data on the association 
between BET-family bromodomain epigenetic reader proteins gene
transcription control and heart failure. This review demonstrates
the importance of the BET proteins epigenetic transcriptional 
control in cardiac remodeling.

Author Response

Thank you for your comment. 

Reviewer 4 Report

This review outlines the roles of BET protein and its inhibitor. There are many previous reviews about BET protein, such as Arun P, et al. J Physiol. 2020;598:3005-3014., Saptarsi M. H, et al. J Mol Cell Cardiol. 2014;74:98-102., and this review does not add much novelty. Therefore, the authors should describe other proteins, including bromo domains such as BRG1 and proteins related to Brd4 in heart failure. For the above reasons, it is my view that this paper is not fit for publication.­­

Major comment;

  1. The authors’ review describes the relationship between Brd4 and histone acetylation, but it does not clearly describe the importance of histone acetylation during heart failure. This should be added. In addition, there are many proteins that contain bromo domains, such as Brd4. The authors should describe the role of such proteins in heart failure.

  1. Brd4 rarely functions alone in heart failure and often works in concert with other factors. Therefore, the authors should describe the proteins related to Brd4 that are involved in heart failure.

Author Response

Reviewer 4

Comments and Suggestions for Authors

This review outlines the roles of BET protein and its inhibitor. There are many previous reviews about BET protein, such as Arun P, et al. J Physiol. 2020;598:3005-3014., Saptarsi M. H, et al. J Mol Cell Cardiol. 2014;74:98-102., and this review does not add much novelty. Therefore, the authors should describe other proteins, including bromo domains such as BRG1 and proteins related to Brd4 in heart failure. For the above reasons, it is my view that this paper is not fit for publication.­­

The reviewer is correct that there are now quite a few review articles on BETs. This is a rather active area of biomedical research given the advent of a large number of BET inhibitors that are now clinically available in the last decade. This fact highlights the  relevance and impact of our manuscript. However, we respectfully disagree that ours does not add novelty. First, there are fewer reviews on heart failure and cardiovascular disease; the vast majority of BET reviews address their role in cancer biology and treatment. As evidence of this, the 2 cited articles do not cover the most recent information on BETs in HF, which our article does. Specifically, the 2014 article by Dr. Haldar only covers the initial 2 papers published in 2013 that described the effects of BET inhibition in cardiac hypertrophy models. Further, the review by Drs. Padmanabhan and Haldar in J Physiology was actually published in final format online in early 2019 and is a text version of their presentation at a Gordon conference in 2018. Hence, this (most current) comprehensive review of BETs in the heart does NOT cover the 4 critical papers published in 2020, all in high impact journals. Our article integrates those earlier results with these 4 new papers, representing a doubling of the major papers in the field of BETs specifically in myocardial pathobiology, and thereby providing a comprehensive and contemporary review of the present understanding of BET biology in the heart.

Major comment;

  1. The authors’ review describes the relationship between Brd4 and histone acetylation, but it does not clearly describe the importance of histone acetylation during heart failure. This should be added. In addition, there are many proteins that contain bromo domains, such as Brd4. The authors should describe the role of such proteins in heart failure.

  1. Brd4 rarely functions alone in heart failure and often works in concert with other factors. Therefore, the authors should describe the proteins related to Brd4 that are involved in heart failure.
  1. Thank you for this suggestion. We have added a paragraph on the importance of histone acetylation on gene expression in HF (lines 80 to 102). This is a nice segue into the mechanisms by which histone acetylation is translated into actual changes in transcription (i.e., the role of BETs)!

While there are other proteins in the heart that have bromodomains (i.e. CBP/p300, TRIM24, multiple components of the SWI/SNF complex, etc…), some of which may have  a role in HF, very little is known specifically about the function of the BD itself in these non-BET proteins, at least in cardiac biology. Further, none show any meaningful inhibition with BET inhibitors (see fig 1b in Filippakopoulos Nature 2010). Thus, off-target effects when studying BET inhibition are not anticipated. Because of this lack of data regarding BD function in non-BET proteins in the heart and the lack of expected cross-interactions in BET inhibitor studies, we have not added another section. Perhaps the reviewer could be more specific on what other non-BET proteins they feel would fit nicely with this discussion.

  1. We agree and this is a very active area of study. We have covered this topic specifically relating to the outstanding new data from the Padmanabhan lab (see section 7) as well as reviewing the well-established interaction between BRD4 and P-TEFb (see sections 1 & 2).